# Effectiveness of Intramammary Antibiotics, Internal Teat Sealants, or Both at Dry-Off in Dairy Cows: Milk Production and Somatic Cell Count Outcomes

**DOI:** 10.3390/vetsci9100559

**Published:** 2022-10-11

**Authors:** Wagdy R. ElAshmawy, Emmanuel Okello, Deniece R. Williams, Randall J. Anderson, Betsy Karle, Terry W. Lehenbauer, Sharif S. Aly

**Affiliations:** 1Veterinary Medicine Teaching and Research Center, School of Veterinary Medicine, University of California Davis, Tulare, CA 93274, USA; 2Department of Internal Medicine and Infectious Diseases, Faculty of Veterinary Medicine, Cairo University, Giza 12613, Egypt; 3Department of Population Health & Reproduction, School of Veterinary Medicine, University of California Davis, Davis, CA 95616, USA; 4California Department of Food and Agriculture, Animal Health Branch, Sacramento, CA 95814, USA; 5Cooperative Extension, Division of Agriculture and Natural Resources, University of California, Orland, CA 95963, USA

**Keywords:** DCT, teat sealants, intramammary antibiotics, somatic cell count, milk, dairy cows

## Abstract

**Simple Summary:**

A block randomized trial was conducted in California dairies to evaluate the effect of different treatments at dry-off on the subsequent lactation’s milk production and somatic cell count (SCC). Greater milk production was observed in cows treated with antibiotics and teat sealants at dry-off in comparison to controls, while there was no significant difference in the milk produced by cows that received either antibiotic or teat sealant in comparison to the untreated cows. Different dry cow treatments were associated with a reduction in ln SCC during the first 150 DIM, with the greatest reduction associated with both antibiotic and teat sealant in comparison to controls. Dairies with high SCC may benefit from treating cows at dry-off with AB, TS, or both.

**Abstract:**

Mastitis is the greatest disease challenge for dairy producers, with substantial economic impacts due to lost milk production. Amongst the approaches implemented to control and prevent mastitis on dairies are vaccination, pre- and post-milking teat dips, and treatments at dry-off including intramammary antibiotics and teat sealants. The objectives of our study were to evaluate the effect of different treatments at dry-off on the subsequent lactation’s milk production and somatic cell count (SCC). A single-blinded controlled block randomized clinical trial was conducted between December 2016 and August 2018 on eight herds from four of the top ten milk-producing counties in California: Tulare, Kings, Stanislaus, and San Joaquin. The trial was repeated with cows enrolled during the winter and summer seasons to account for seasonal variability. Eligible cows were treated at dry-off with either intramammary antibiotics (AB), internal teat sealant (TS), AB + TS, or did not receive any treatment (control), and were followed through 150 days in milk (DIM) post-calving. The milk production and SCC data were extracted from monthly test day milk records (Dairy Comp 305, Valley Ag Software, Tulare, CA, USA). Two-piece spline linear mixed models were used to model the milk production (kg) and natural logarithm-transformed SCC. After accounting for parity, breed, season, and dry period duration, the milk model showed a significant increase in milk production (1.84 kg/day) in cows treated with AB + TS at dry-off in comparison to controls. There was no significant difference in the milk produced by cows that received either AB or TS (0.12 kg/day, and 0.67 kg/day, respectively) in comparison to the untreated cows. Different dry cow treatments were associated with a significant reduction in ln SCC during the first 150 DIM. The greatest reduction was associated with using AB + TS, followed by AB, and finally TS in comparison to controls. Dairies with high SCC may benefit from treating cows at dry-off with AB, TS, or both.

## 1. Introduction

Mastitis continues to challenge dairy producers due to the severe economic losses associated with reduced productivity, increased risk of early culling, treatment and labor costs, and milk discarded during treatment [1,2,3,4]. Strategies used to prevent and control mastitis include the administration of vaccines against coliform bacteria and *Staphylococcus aureus*, the application of pre- and post-milking teat dips, and the application of dry cow therapy (DCT) [5,6,7,8]. Dry cow therapy is one of the five points of the Mastitis Control Plan recommended since 1970 [9,10]. Such treatment may resolve clinical and subclinical mastitis that exists at dry-off, reducing the risk of mastitis during the dry period and after calving [11,12,13]. As a result, the dairy industry has broadly adopted the use of intramammary infusion with a long-acting antibiotic at dry-off, a practice known as blanket dry cow therapy (BDCT). In the US, approximately 93% of dairy cows are treated with intramammary antibiotics at dry-off, representing approximately 80.3% of the surveyed US dairy herds [7]. Approximately 36.9% of dairies in the US use internal teat sealants at dry-off [7]. Teat sealants have the advantage of acting as a physical barrier that hinders the entrance of mastitis-causing microorganisms into the teat canal. Recent studies revealed that not all cows at dry-off benefit from intramammary antibiotic treatment [14,15,16,17,18,19,20]. The latter studies concluded that dry cow therapy should only target dairy cows at high risk of mastitis to reduce the unnecessary use of antibiotics at dry-off in a practice called selective dry cow therapy (SDCT).

Antibiotics are valuable chemicals that safeguard the health and welfare of humans and animals; however, their misuse leads to the development of resistant bacterial strains [21]. A US study of antimicrobial use in dairy cows based on grams of active substance per cow year showed approximately nine times greater use for dry cow therapy compared to use for clinical mastitis [22]. Hence, SDCT has the potential to result in a greater reduction in antimicrobial drug use compared to stewardship efforts related to other diseases. Furthermore, previous studies found no differences in the SCC and milk production between cows that received SDCT or BDCT in the subsequent lactation [16,17,18,20,23]. The aforementioned studies were limited to low-risk herds with a history of low bulk tank somatic cell count (BTSCC) and/or low-risk cows with SCC < 200,000 cells/mL, which may not reflect the current situation for the majority of the dairy herds; additionally, none of them included a non-treated control group [16,17,18,20,23]. Coupled with the inclusion of herds across the range of bulk tank SCC, a non-treated control group offers estimates for mastitis, subclinical mastitis, milk production, and culling baseline risk—a necessary input for future studies developing and validating SDCT algorithms and their economics.

The current study hypothesis was that dry cow treatments (AB, TS, and AB + TS) affect milk production and SCC in the subsequent lactation. Our objectives were to estimate the effect of using different dry cow treatments (AB, TS, and AB + TS) on milk production and SCC during the first 150 DIM of the following lactation across the seasonal variation on California dairies.

## 2. Materials and Methods

### 2.1. Herd Selection

The current research was approved by the University of California Davis Institutional Animal Care and Use Committee (protocol number 19761). A single-blinded controlled block randomized clinical trial was conducted between 28 December 2016 and 1 August 2018. Eight dairy herds were enrolled in the study from four of the top ten milk-producing counties in California [24]. Three herds were in Tulare County, two in Kings County, two in Stanislaus County, and one in San Joaquin County. The study commenced with an in-person enrollment survey. The cows were enrolled between December 2016 and March 2017 (winter season) and between June 2017 and September 2017 (summer season), with all eight herds sampled during both seasons. The herd demographics and survey information are previously described [25]. Briefly, the study dairies had a mean lactating herd size of 1782 cows, which were primarily Holstein, Jersey, or crossbreeds. The study herds had a wide range of BTSCC (<200,000 to 400,000 cells/mL) and an average of 305,571 cells/mL.

### 2.2. Cow Enrollment, Follow-Up, and Sample and Data Collection

The cows were enrolled on a weekly basis from the study herds and every two weeks from one herd according to the dairies’ dry-off schedules until the study sample size was achieved. The inclusion and exclusion criteria are previously described [25]. Briefly, all cows presented for dry-off were inspected by the study personnel at the parlor entry and any cows with low BCS or lameness were excluded [26]. Prior to milking, the cows were visually inspected and excluded if any signs of clinical mastitis (swelling, redness, and painful udder) were observed. In addition, the udder hygiene score (scale 1–4), teat-end score (scale 1–4), and California Mastitis Test (CMT; score 0, trace, 1–3) were recorded for each enrolled cow. After milking, the enrolled cows were block randomized, with each cow allocated one of the four treatments: intramammary antibiotics (AB), internal teat sealant (TS), both (AB + TS), or no treatment (None). The enrolled cows were followed up until 150 DIM post-calving. The udder hygiene, teat-end, and CMT scores were again determined for each cow prior to the post-calving milk sample collection [27,28,29,30]. At both times, the study personnel recording udder hygiene, teat-end, and CMT scores were blinded to the cows’ treatment allocations. Milk production and SCC were recorded through the Dairy Herd Improvement Association (DHIA) monthly herd testing using Dairy Comp 305 (Valley Ag Software, Tulare, CA, USA) for all study herds.

### 2.3. Statistical Analyses

The effects of intramammary treatments on milk production and SCC (outcomes) during the first 150 DIM in the subsequent lactation were modeled using two-piece spline general linear mixed models. In addition to treatment, the season (winter vs summer), cow-related variables including cow breed (Holstein, Jersey, crossbreed), parity (2nd, ≥3rd lactation), history of clinical mastitis, and SCC were tested in the model. Other explanatory variables recorded during the enrollment and after calving, such as udder hygiene score, teat-end score, and CMT score, were included in the model.

#### 2.3.1. Modeling Milk Production during the First 150 DIM after Calving

A two-piece splines general linear mixed model (LMM) was used to model the effect of different treatments on milk production during the first 150 DIM in the lactation following the enrollment. Splines have been used to model milk production resulting in pre- and post-peak linear predictions that meet at a fixed knot representing DIM at peak milk production [31,32]. Subsequent modifications allowed for a lactation-specific knot such that the two splines meet at the naturally observed DIM at peak milk production for each lactation [33] and have been applied to other dairy cattle health outcomes [34].

Equation (1) summarizes the model used to estimate the regression coefficients for the association between different treatments (AB, TS, or AB + TS, versus None) and the outcome ($y_{jkl}$) milk production during the first 150 DIM of the following lactation.
(1)$$\begin{array}{c} y_{jkl}=\beta_{0}+\beta_{1}X_{1}+{t^{dairy}}_{l}+u{\;}_{kl}^{block}+v_{0}{\;}_{jkl}^{cow}{}_{\;}\\ \;\;\;\;\;+v_{1}{\;}_{jkl}^{cow}DIMprePk_{jkl}+v_{2}{\;}_{jkl}^{cow}DIMpostPk_{jkl}+e_{jkl} \end{array}$$

##### Random Effects

The random effects included *t, u*, and *v* representing the random intercepts for dairy, block, and cow, respectively, where *j* cows were nested within randomization block *k*, which were in turn nested within dairy *l*. The peak test-day milk production was determined for each cow’s lactation as the DIM at the greatest test-day milk production recorded in the 150 DIM after calving. Two random slope coefficients were specified at the cow level for repeated milk measures during the lactation pre- and post-peak test-day milk production (splines), $v_{1}{\;}_{jkl}^{cow}DIMprePk_{jkl}$, and $v_{2}{\;}_{jkl}^{cow}DIMpostPk_{jkl}$, respectively. The formula for the calculation of pre-peak test-day DIM is DIMprePk = (DIM at test-day–DIM at peak)/(DIM at peak), while the formula for post-peak test day DIM is DIMpostPk = (DIM at test-day − DIM at peak)/(total DIM − DIM at peak). All random effects ($t_{l},\;u_{k},\;v_{j}$) and residual errors ($e_{jkl}$) were assumed to be normally distributed with mean = 0 and variances of σ^2^_t_, σ^2^_u_, σ^2^_v_, and σ^2^, respectively.

##### Fixed-Effect Variables

The intercept ($\beta_{0}$) represents the mean volume of milk produced on the peak test-day, the estimate of which can be interpreted as the mean peak test-day milk produced across all of the study cows. Similar to the random effects, fixed effects also include DIMPrePk and DIMPostPk, which are the splines (lactation shape parameters) that meet at the peak test-day and were calculated as described above. The term $\beta_{1}X_{1}$ represents the fixed-effects variables in the model including the treatment groups (AB, TS, AB + TS, versus None) amongst other explanatory variables. Cow-related factors obtained from the cow records including breed (Jersey, Holstein, Cross), parity (second lactation, ≥3 lactation), history of clinical mastitis events prior to enrollment, the time elapsed between the last mastitis event prior to enrollment and enrollment date (cows with no history of mastitis were assigned the duration of their lactation), DIM at enrollment, and the length of the dry period were included in the model. Clinical mastitis events prior to enrollment were specified as four variables; clinical mastitis during the enrollment lactation (Yes/No), the number of clinical mastitis events during enrollment lactation, clinical mastitis prior to the enrollment lactation (Yes/No), and the number of mastitis events that occurred during lactations prior to enrollment lactation for multiparous cows. Other factors recorded at enrollment included season, udder hygiene score (1–4), teat-end score (1–4), and CMT score (0, T, 1, 2, 3). The udder hygiene score was presented to the model as a four-level variable (scores 1, 2, 3, 4) and two-level variable (≤2, >2). At the cow level, the teat-end score was coded using the teat with the highest score. In addition, the teat-end score was also explored at the quarter level. Specifically, six different variables were used for coding the teat-end score: cow with teat-end score at any teat ≥2 (Yes/No) and the number of teats with teat end score ≥2/cow; cow with teat-end score at any teat ≥3 (Yes/No) and the number of teats with teat-end score ≥3/cow; and cow with teat-end score at any teat ≥4 (Yes/No) and the number of teats with teat-end score ≥4/cow. Similarly, the CMT score was coded at the cow level assigned the highest quarter score. In addition, eight different variables were used to specify the CMT score at the quarter level: cow with CMT at any quarter ≥ trace (Yes/No) and the number of quarters with score ≥ trace/cow; cow with CMT score at any quarter ≥1 (Yes/No) and the number of quarters with CMT score at any quarter ≥2/cow; and cow with CMT score at any quarter =3 (Yes/No) and the number quarters with CMT score = 3/cow. The length of the dry-off period was explored as a continuous variable and as a categorical variable with three levels (0–60, 61–120, >120 days) based on its distribution. The last SCC test before enrollment was explored as a continuous untransformed variable and as a natural log transformation. In addition, the SCC of the last one, two, three, four, five, or six DHIA tests before enrollment were introduced to the model individually as categorical variables (Yes/No) indicating whether any of these DHIA tests exceeded the cut-off of ≥200, ≥250, ≥300, ≥350, ≥400, ≥450, or ≥500 ×1000 cells/mL. Similarly, SCC was also specified as the number of tests in the last one, two, three, four, five, or six DHIA tests before enrollment that exceeded the previous cut-offs. Hence, the two categorical SCC variables informed the model as to whether SCC exceeded the previous cut-offs and the number of tests where SCC exceeded the previous cut-offs over the six DHIA tests prior to dry-off. Each variable was included in univariable models with the random-effect variables described above. Significant variables in the univariable models were fitted in the final model, and variables that were no longer significant were excluded from the final model.

#### 2.3.2. Modeling Somatic Cell Count during 150 DIM after Calving

Similar to the milk production model (Equation (1)) described above, a two-piece splines general LMM was used to model the effect of different treatments on the natural logarithm of SCC expressed in cell/mL during the first 150 DIM in the lactation following enrollment [33].

##### Random Effects

The random-effect variables included in the SCC model were specified similar to those in the milk model, with the exception of the following differences:

The minimum (Min) test-day SCC was determined as the lowest SCC measure recorded in the 150 DIM after calving. The formula for the calculation of pre-Min test-day SCC DIM is DIMpreMin = (DIM at test date − IM at Min)/(DIM at Min), while the formula for test days post-Min is DIMpostMin = (DIM at test date − DIM at Min)/(total DIM − DIM at Min). All random effects ($t_{l},\;u_{kl},\;v_{jkl}$) and residual errors ($e_{jkl}$) were assumed to be normally distributed with mean = 0 and variances of σ^2^_t_, σ^2^_u_, σ^2^_v_, and σ^2^, respectively.

##### Fixed-Effect Variables

The explanatory variables for the SCC model were specified as in the milk model described above. In addition, milk production (Kg) during the testing lactation was offered to the model as a continuous variable to account for the variation in the SCC that may be attributed to the volume of milk produced. A log-transformed SCC of the test prior to enrollment was included in the model after accounting for the duration between the test date and enrollment date.

#### 2.3.3. Selection of the Final Models

For both the milk and SCC models, univariable models were fitted for each variable along with the respective random effect structure described above. Model building was performed manually by including the spline fixed effects, treatment, and the remaining fixed effects described above for each model. A manual backward elimination process was implemented, and a 5% level of significance was used in all models. Confounding by known confounders was assessed during variable selection and model building, using a 20% change in estimates method, and two-way interactions for potential effect modifiers were tested using significance testing [33]. The model goodness of fit was estimated using the Akaike Information Criterion (AIC) to select between competing models, with lower values denoting a better model fit [35]. The addition of a quadratic term for each outcome’s respective spline terms (*DIMprePK squared* and *DIMpostPK squared* for the milk model, and *DIMpreMin squared* and *DIMpostMin squared* for the SCC model) was explored in terms of improving the model fit as assessed by a lower AIC compared to the models without these quadratic term variables. Locally weighted scatterplot smoothing (LOWESS) was used to plot predictions (Xβ) for each of the outcomes, milk production, and SCC, by parity (second, and greater than second lactation) to compare the study treatments. The statistical analysis and visualizations were conducted in Stata IC 15.1 (College Station, TX, USA).

## 3. Results

### 3.1. Description of the Enrolled Herds

A total of 1133 cows were enrolled in the study (480 in winter, 653 in summer), with 27 being excluded due to errors transcribing their identification, resulting in a total of 1106 cows. Of the 1106 cows, 45 cows were culled during the dry period and another 166 cows were culled during the first 150 days post-calving (89 of the 166 were culled between calving and the first DHIA test). The herd demographics have been described elsewhere (Aly et al., 2022). Briefly, the average number of lactating cows in the enrolled herds was 1782 (SE ± 347) milking cows. The cow breeds included in the study were Holstein only (four herds), Holstein, Jersey, and crossbreds (two herds), Holstein and Jersey (one herd), and Jersey and crossbreed (one herd). All enrolled herds used Dairy Comp 305 (Valley Ag Software, Tulare, CA, USA) as DHIA computer software. The breed and parity of the cows included in the analyses are summarized in Table 1. Table 2 summarizes the mean test-day milk production and mean test-day SCC by treatment group in the study cows.

### 3.2. Milk Production

Table 3 summarizes the effect of the different dry cow treatments (AB, TS, AB + TS, None) on milk production during the first 150 DIM using two competing models. The first (Model A) modeled milk production as explained by the set of predictors that maximized the model fit (AIC, 39,811), which included test-day SCC prior to dry-off. The second model (Model B) was specified without SCC, since not all US dairy herds may have this information readily available (AIC, 39,862). Instead, the CMT score at dry-off was forced in Model B to offer an alternative for producers who may not subscribe to herd SCC testing. Figure 1 depicts the lactation-specific LOWESS plots for milk production using Model A’s fixed effects, which included SCC given its better fit compared to Model B that substituted SCC for CMT.

There was a significant increase in the daily milk production (Model A 1.84 kg/cow/day, Model B 1.96 kg/cow/day) for cows that received both intramammary antibiotics and internal teat sealants at dry-off in comparison to the control group (*p* < 0.01). In contrast, there was no significant difference in milk production between the untreated cows and cows that received intramammary antibiotic tubes (Model A 0.12 kg/cow/day, *p* = 0.83 and Model B 0.17 kg/cow/day, *p* = 0.76) or internal teat sealants (Model A 0.67 kg/cow/day, *p* = 0.24 and Model B 0.69 kg/cow/day, *p* = 0.23), although the numerical difference in the latter comparison may be biologically important. The Bonferroni-adjusted multiple comparisons showed no additional significant differences, with the exception of the difference between cows that received both antibiotics and internal teat sealants compared to those that received antibiotics only (Model A, *p* = 0.02, Model B, *p* = 0.01). There was a significant difference in the milk production between parities (*p* < 0.01), with cows of third or greater lactation producing more milk (Model A 2.28 kg/cow/day, Model B 2.15 kg/cow/day) compared to second lactation cows. The SCC-based model (Model A) showed a significant decrease in the daily milk production of cows with SCC ≥ 200,000 cells/mL at any test during the enrollment lactation (1.00 kg, *p* = 0.03), in comparison to cows with <200,000 cells/mL. The CMT-based model (Model B) revealed that cows with a score of two or more in one or more quarters at enrollment had significantly lower daily milk production (1.03 kg/cow/day, *p* = 0.04) in comparison to cows with a CMT score less than two in all quarters. Both models showed that the cows enrolled during the summer season produced significantly less daily milk (Model A 4.61 kg/cow/day, Model B 4.51 kg/cow/day) compared to cows enrolled during the winter season (*p* < 0.01). Both models showed that every one-day increase in the dry period was associated with a daily increase of 0.03 kg of milk (*p* 0.03 and 0.01, respectively). For Model A, each one-day increase in the duration between the last clinical mastitis event and dry-off was associated with a significant increase of 0.007 kg daily milk (*p* = 0.01). In addition, a model with a quadratic term for dry period length showed minimal improvement in AIC (Model A with the variable dry period length as a main effect had an AIC = 39,811 versus the same model with the addition of the quadratic term for dry period length, which had an AIC = 39,805), with approximately similar estimates for the remaining variables and no change in their interpretations; hence, the model with the main effect only was chosen for simplicity.

### 3.3. Somatic Cell Count

The effect of dry cow treatments (AB, TS, AB + TS, None) on ln SCC during the first 150 DIM following calving is shown in Table 4. Figure 2 depicts the lactation-specific LOWESS plots using fixed effects from the final model for SCC.

Different dry cow treatments were associated with a significant reduction in the ln SCC during the first 150 DIM following calving. The highest reduction was associated with cows that received both intramammary antibiotics and internal teat sealants, followed by cows that received intramammary antibiotics only, and cows that received internal teat sealant only, in comparison to the untreated cows in the control group. The Bonferroni-adjusted multiple comparisons showed no additional significant differences between the treatment groups. In comparison to the Holsteins, the Jerseys had significantly lower ln SCC, while crossbreeds had a non-significant (*p* = 0.08), but numerically lower ln SCC. Cows of third lactation or greater had a significantly higher ln SCC compared to cows in their second lactation. Cows with a teat-end score of four on any teat after calving had a higher ln SCC compared to cows with a teat-end score less than four. Cows with a CMT score of three in any quarter after calving had higher ln SCC compared to cows with a CMT score less than three. Cows with a history of clinical mastitis during the enrollment lactation or any lactation prior to enrollment had significantly higher ln SCC compared to cows with a history of no clinical mastitis. Each additional kilogram of daily milk produced was associated with a decrease in the ln SCC. Model coefficients are on the natural logarithm scale, and hence estimates comparing cows should incorporate the intercept and covariate coefficients relevant to the contrast. The example below compares two Holstein cows with 100,000 SCC/mL 30 days prior to dry-off and not treated with any dry-off therapy. In addition, the first cow completed her first lactation at dry-off, while the second cow completed her second lactation at dry-off with at least one teat-end score of 4, at least one quarter with a CMT score of 3, and a history of mastitis in both previous lactations. Assuming both cows are at 40 DIM in the lactation subsequent to their dry-off with a SCC nadir at 60 DIM, then the estimates based on the SCC model are 119,372 cells/mL and 1024,791 cells/mL for the first and second cows, respectively.

To further demonstrate the differences in SCC due to treatment under the same conditions, 40 DIM in the subsequent lactation with a SCC nadir at 60 DIM, had the first cow been treated with AB, her estimated SCC would have been 88,432 cells/mL. Similarly, if the first cow under the same conditions had been treated with TS or AB + TS, her SCC would have been 98,715 cells/mL or 79,221 cells/mL, respectively. The second cow’s estimated SCC after treatment with AB, TS, or AB + TS at dry-off, assuming the same conditions, would have been 759,184 cells/mL, 847,460 cells/mL, or 680,103 cells/mL, respectively.

## 4. Discussion

Different dry cow therapy practices including the administration of intramammary antibiotics, internal teat sealants, or both were associated with increased milk production; however, the increase was only significant for cows that received both intramammary antibiotics and internal teat sealant. There was a significant reduction in the ln SCC in all treated groups (AB, TS, and AB + TS) in comparison to the control group that did not receive any treatment. Dry cow treatments are likely to reduce the prevalence of intramammary infection at the commencement of the subsequent lactation, either by curing existing infections and/or reducing the risk of new infection over the dry period [36]. The net effect, therefore, is likely to be that dry cow treatments reduce the prevalence of intramammary infection at calving, which may potentially impact production and SCC. Both milk production and ln SCC were significantly associated with parity, breed, and season. Cows enrolled during the summer season had a significant reduction in both milk production and SCC.

The study revealed a significant increase in milk production during the first 150 DIM in cows treated with both intramammary antibiotics and internal teat sealant in comparison to cows that did not receive any treatments at dry-off. In addition, there was a non-significant increase in the milk production of cows that received either internal teat sealants or intramammary antibiotics at dry-off—specifically, cows treated with TS produced numerically more milk than those treated with AB. Based on Models A and B for milk production, cows with at least one high test-day SCC (>200,000 cells/mL) or CMT score of 3 in any quarter during the enrollment lactation can benefit from combined treatment with AB + TS. The study findings agree with findings by McNab and Meek (1991), who reported an increase in milk production in cows that received dry cow treatment compared to non-treated cows. In addition, Osteras and Sandvik (1996) found a significant increase in the milk production between treated and non-treated groups. The current study findings are in contrast with findings from previous studies that reported no differences in the milk production between cows that received SDCT and BDCT [16,19,37,38]. The lack of difference in findings reported by Rajala-Schultz et al. [36], Cameron et al. [19] and Vasquez et al. [16] is likely attributed to the assignment of the enrolled cows into SDCT and BDCT groups according to the history of clinical mastitis and SCC in the enrollment lactation; in contrast, our study cows were randomized to different treatment groups regardless of their clinical mastitis history. Furthermore, our study had an untreated control group, unlike Cameron et al. [19], Golder et al. [37], Vasquez et al. [16] and Rowe et al. [17]. In addition, our analysis relied on use of two-piece spline mixed regression models, which account for the pre-peak and post-peak differences in the milk production, while Rajala-Schultz et al. [36], Cameron et al. [19], Golder et al. [37], and Vasquez et al. [16] used linear mixed regression models, which assume a linear relation between milk production and days in milk Vasquez et al. [16] only followed the milk production for the first 30 days in milk, compared to our study, in which the cows were followed up to 150 DIM. did not include internal teat sealants in their study, while in our study, the cows received teat sealants either alone or in combination with intramammary antibiotics. Rowe et al. [17] used linear mixed models in their analysis for milk and ln SCC models with DIM categorized into six levels equally spanning the first 120 DIM, which assumes a flat relationship between the respective outcome and DIM within each category and hence ignores the non-linear trends in milk production and SCC. Such differences in the study design, enrollment criteria, and analysis models may explain the differences regarding the effect of dry cow treatment on the milk production between studies.

There was a significant decrease in ln SCC during the first 150 DIM for the cows treated with either AB, TS, or both in comparison to the cows that did not receive any treatment at dry-off. The cows that received both AB and TS had the lowest SCC, followed by those treated with either AB or TS alone. The cows with a teat-end score of four on any teat, a CMT score of three in at least one quarter at dry-off, or mastitis in the enrollment or previous lactations may benefit from lowering their SCC by treatment with AB or TS at dry-off, with the maximum benefit derived when treated using both. Our results agreed with the findings of Golder et al. [37], as they found that treatment with a combination of teat sealants and intramammary antibiotics at dry-off was associated with a lower geometric mean of SCC in comparison to antibiotics alone. Rajala-Schultz et al. [36] found a 16% decrease in SCC in cows treated with intramammary antibiotics in comparison to the non-treated group. McDougall [39] reported a significant decrease in the SCC of cows treated with antibiotics and teat sealants at dry-off in comparison to a non-treated group [39]. The application of dry cow therapy to all cows was associated with a significant reduction in SCC compared to no treatment [39]. A previous study reported a significant reduction of 0.409 ln unit on the geometric mean of SCC (1000/mL) in cows treated with antibiotic therapy at dry-off in comparison to non-treated groups in the lactation following enrollment, and they recommended the use of SDCT [40]. In our study, we used a two-piece mixed regression model to account for the dip in the SCC curve, a biological feature that coincides with the peak milk production, while Cameron et al. [19] and Vasquez et al. [16] used a linear mixed regression model for their analysis, which did not account for the dip in the SCC curve. Mutze et al. [41] reported a significant decrease in the SCC after calving in comparison to the last test before dry-off, but there was no significant difference in the SCC between cows that received intramammary antibiotics at dry-off and cows that received both intramammary antibiotics and internal teat sealants [41].

The findings from the current study show that cow breed, parity, and season were significant factors that affected both milk production and SCC during the lactation. Our results disagree with [42,43], as they reported higher SCC for Jersey cows compared to Holsteins, while [44,45] found no difference in the SCC between Holstein and Jersey breeds. Cows of the third or greater lactation produced more milk and had higher SCC in comparison to second lactation cows, similar to the results of previous studies [41,46]. Our results disagreed with Cameron et al. [19], as they reported a decrease in milk production for cows in the third or greater lactation in comparison to second lactation cows, which could be related to their inclusion of previous lactation milk production in their model, which is on the causal pathway between parity and milk production. The cows enrolled during the summer season produced less milk than those enrolled during the winter season, which could be due to heat stress, which can decrease feed consumption and indirectly reduce the milk yield [44].

Milk production and SCC in the first 150 DIM were associated with SCC of the enrollment lactation. Cows with SCC ≥ 200,000 cell/mL at any test during the enrollment lactation produced less milk in comparison to cows with <200,000 cells/mL. The lower milk production in cows with higher enrollment lactation SCC could be attributed to the higher risk of existing intramammary infection, which may continue to the following lactation. In addition, a higher SCC at enrollment could be a residual effect of clinical mastitis during the enrollment lactation, and these cows may be of higher risk for the development of clinical mastitis during subsequent lactations. The current study showed that cows with a CMT score of three in at least one quarter at enrollment produced lower milk and higher ln SCC in the following lactation in comparison to cows with lower CMT scores. A CMT-positive test (score 3) is an indication of an intramammary infection of the affected quarter. Hence, cows with higher CMT scores at enrollment may suffer from subclinical mastitis and have a higher risk of developing clinical mastitis during the next lactation. Despite the lack of effect modification (interaction) between the treatment and cow characteristics in both milk production and SCC models, selective dry cow therapy can still benefit cows at risk of reduced milk production or elevated SCC. As a result, economics and antimicrobial drug stewardship practices may further help determine which cows may benefit from dry cow therapy.

## 5. Limitations

The study included a large number of cows from four of the top ten milk-producing counties in California (Tulare, Kings, Stanislaus, and San Joaquin) during the summer and winter seasons, but the weather conditions and cow housing may differ from other regions. Hence, similar studies may be needed to address other climatic and housing conditions. The study included Holstein, Jersey, and crossbreeds as they were the common milk-producing breeds in California; however, breeds may also differ in other countries. The study relied on the monthly milk test for milk production and SCC data, as this is the common practice in California, so further research may investigate daily milk and SCC where and when such data are available. Information on the microbiological quality of the milk could also further inform models predicting milk and SCC; however, such variables are likely highly correlated with CMT scores and SCC, and are on the causal pathway between the treatment and the study outcomes. Future studies may be required to evaluate the effect of different dry cow therapies on the bacteriological quality of milk. Finally, although the inclusion of different dry cow antibiotics in our trial increases its generalizability compared to the use of a single antibiotic, estimating the effect measure for each drug would be based on a smaller sample size.

## 6. Conclusions

Adjusting for SCC in the previous lactation, the treatment of cows at dry-off with both AB and TS was associated with an increase in the test-day milk production of approximately 1.84 Kg (4.06 lbs) compared to untreated cows. Otherwise, the treatment of cows at dry-off with either AB or TS did not result in statistically significant differences in test-day milk production compared to untreated cows. Dry cow treatments including AB, TS, or both were beneficial in significantly reducing test-day SCC. Cows with high test-day SCC during the enrollment lactation or a CMT score of three in any quarter at dry-off may benefit from combined AB and TS treatments.

## Figures and Tables

**Figure 1 vetsci-09-00559-f001:**
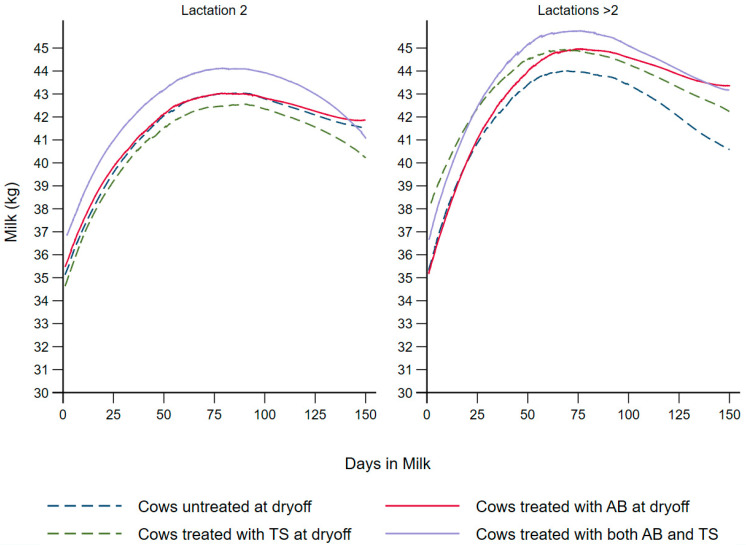
Locally weighted scatterplot smoothing of the predicted milk production curves for dairy cows that received different dry cow treatments (AB, TS, AB + TS or None) during the first 150 days in milk post-calving. AB: cows treated with intramammary antibiotics at dry-off. TS: cows treated with internal teat sealants at dry-off. AB + TS: cows treated with both intramammary antibiotics and internal teat sealants at dry-off. None: control group.

**Figure 2 vetsci-09-00559-f002:**
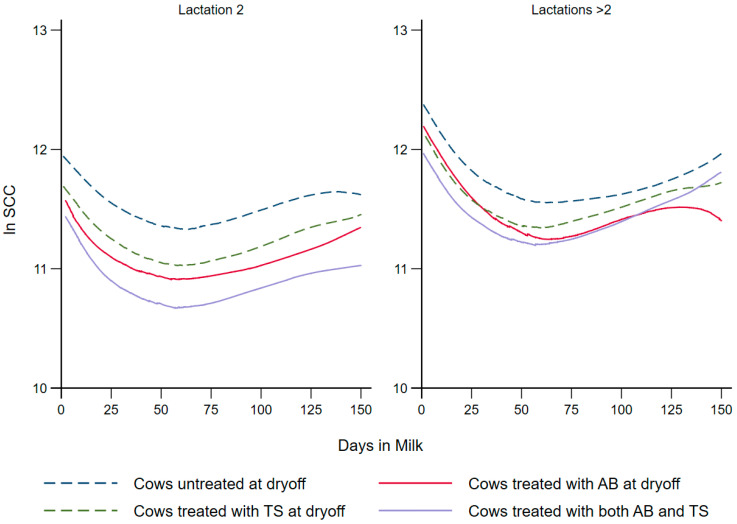
Locally weighted scatterplot smoothing of the predicted natural log of somatic cell count curves for dairy cows that received different dry cow treatments (AB, TS, AB + TS or None).

**Table 1 vetsci-09-00559-t001:** Comparison of the categorical baseline traits for dairy cows (*n* = 972 cows) from eight herds block randomized to one of four treatments at dry-off (AB, TS, AB + TS, None) *.

Parameter	Level	Treatment					*p* **
		None	AB	TS	AB + TS	Total	
Number of cows		252	244	228	248	972	0.365
Breed	Holstein	165	157	147	153	622	0.85
	Jersey	49	47	43	47	186	
	Cross	38	40	38	48	164	
Parity	2	113	104	95	122	434	0.85
	≥3	139	140	133	126	538	

* AB, antibiotic; TS, internal teat sealants. ** Chi-square test.

**Table 2 vetsci-09-00559-t002:** Mean test-day milk production and mean test-day SCC during the first 150 DIM from eight herds block randomized to one of four treatments at dry-off (AB, TS, AB + TS, None) *.

Treatment Group	Test-Day Milk (kg)			Test-Day SCC (1000 Cells/mL)		
	Mean	SE	95% CI	Mean	SE	95% CI
None	41.73	0.288	(41.16–42.29)	330.22	19.03	(292.90–367.55)
AB	42.36	0.289	(41.79–42.93)	264.97	19.12	(227.48–302.46)
TS	42.28	0.304	(41.68–42.88)	302.32	20.13	(262.86–341.79)
AB + TS	43.30	0.287	(42.74–43.87)	245.75	19.00	(208.49–283.01)

*AB, antibiotic; TS, internal teat sealants.

**Table 3 vetsci-09-00559-t003:** Final two-piece spline models for test-day milk production (kg) during the first 150 DIM of dairy cows randomized at dry-off to four treatment groups (AB, TS, AB + TS, None) ^1^ on eight California dairies (*n* = 972 cows).

Factor	Levels	Model A ^2^ (AIC; 39,811)			Model B ^3^ (AIC; 39,862)		
		Coefficient (95% CI)	Standard Error	*p*	Coefficient (95% CI)	Standard Error	*p*
Treatment	None	Referent			Referent		
	AB	0.12 (−0.99, 1.23)	0.570	0.83	0.17 (−0.95, 1.29)	0.575	0.76
	TS	0.67 (−0.46, 1.82)	0.583	0.24	0.69 (−0.45, 1.85)	0.588	0.23
	AB + TS	1.84 (0.72, 2.95)	0.569	<0.01	1.96 (0.84, 3.08)	0.573	<0.01
Breed	Holstein	Referent			Referent		
	Jersey	−8.90 (−10.38, −7.42)	0.754	<0.01	−8.94 (−10.43, −7.46)	0.758	<0.01
	Cross	−4.58 (−5.89, −3.26)	0.671	<0.01	−4.53 (−5.85, −3.21)	0.673	<0.01
Parity	Second	Referent			Referent		
	≥3	2.28 (1.34, 3.22)	0.480	<0.01	2.15 (1.22, 3.08)	0.475	<0.01
SCC at any DHIA test during enrollment lactation ≥200,000 cells/ml	No	Referent					
	Yes	−1.00 (−1.92, -0.07)	0.471	0.03			
Cow has at least one quarter with CMT score ≥ 2 at enrollment	No				Referent		
	Yes				−1.03 (−2.04, −0.01)	0.517	0.04
Time between dry-off and last clinical mastitis	Days	0.007 (0.001, 0.01)	0.003	0.01			
Season	Winter	Referent			Referent		
	Summer	−4.61 (−5.55, −3.67)	0.479	<0.01	−4.51 (−5.44, −3.59)	0.472	<0.01
Days dry	days	0.03 (0.003, 0.06)	0.015	0.03	0.03 (0.008, 0.06)	0.015	0.01
*Intercept and splines variables*							
Days in milk pre-peak (Kg)		−0.81 (−3.30, 1.67)	1.270	0.52	−0.74 (−3.24, 1.74)	1.270	0.55
Days in milk post-peak (Kg)		−4.94 (−5.42, −4.45)	0.247	<0.01	−4.95 (−5.44, −4.46)	0.248	<0.01
Days in milk pre-peak (Kg) square		−13.97 (−16.97, −10.98)	1.526	<0.01	−13.91 (−16.90, −10.92)	1.526	<0.01
Days in milk post-peak (Kg) square		−0.30 (−0.44, −0.17)	0.069	<0.01	−0.30 (−0.44, −0.17)	0.069	<0.01
Intercept		45.63 (42.29, 48.97)	1.705	<0.01	47.28 (44.61, 49.94)	1.361	<0.01

^1^ AB, antibiotic; TS, internal teat sealants. ^2^ Model A: model using information about the history of SCC in the enrollment lactation. ^3^ Model B: model using information about CMT score at dry-off (instead of SCC in the current lactation).

**Table 4 vetsci-09-00559-t004:** Final two-piece spline models for natural logarithm-transformed test-day somatic cell counts ^1^ (SCC) during the first 150 DIM of dairy cows randomized at dry-off to one of four treatment groups (AB, TS, AB + TS, None) ^2^ on eight California dairies (*n* = 972 cows).

**Factor**	**Level**	**Coefficient**	**Standard Error**	** *p* **	**95% Confidence Limits**	
					**Lower**	**Upper**
Treatment	None	Referent				
	AB	−0.30	0.086	<0.01	−0.47	−0.13
	TS	−0.19	0.088	0.03	−0.36	−0.01
	AB + TS	−0.41	0.087	<0.01	−0.58	−0.24
Breed	Holstein	Referent				
	Jersey	−0.30	0.111	<0.01	−0.52	−0.08
	Cross	−0.17	0.104	0.08	−0.38	0.02
Parity	Second	Referent				
	≥3	0.21	0.074	<0.01	0.07	0.36
Teat-end score 4 at any teat after calving	No	Referent				
	Yes	0.59	0.167	<0.01	0.26	0.92
CMT 3 at any quarter after calving	No	Referent				
	Yes	0.79	0.138	<0.01	0.52	1.06
Mastitis at enrollment lactation	No	Referent				
	Yes	0.30	0.142	0.03	0.02	0.58
Mastitis at any lactation prior to enrollment lactation	No	Referent				
	Yes	0.26	0.131	0.04	0.008	0.52
Milk production at current lactation	(kg)	−0.01	0.001	<0.01	−0.018	−0.01
Ln SCC of last test before enrollment	Natural log 1000 cells/mL	0.17	0.028	<0.01	0.11	0.23
Time between last test day and enrollment day	days	0.009	0.003	0.01	0.002	0.01
Days in milk pre-Min		−3.69	0.194	<0.01	−4.08	−3.31
Days in milk post-Min		4.28	0.155	<0.01	3.97	4.58
Days in milk pre-Min square		−2.25	0.224	<0.01	−2.69	−1.81
**Factor**	**Level**	**Coefficient**	**Standard Error**	** *p* **	**95% Confidence Limits**	
					**Lower**	**Upper**
Days in milk post-Min square		−3.13	0.148	<0.01	−3.42	−2.84
Intercept		10.05	0.207	<0.01	9.65	10.46

^1^ Model coefficients are on the natural logarithm scale and hence estimates comparing cows should incorporate the intercept and any other covariate coefficients relevant to the contrast. ^2^ AB, antibiotic; TS, internal teat sealant.

## Data Availability

The data presented in this study are available upon reasonable request from the corresponding author. The data are not publicly available, as the study-associated dairy owners did not consent to publishing them alongside the article.

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
