# Peer review of "Effectiveness of Intramammary Antibiotics, Internal Teat Sealants, or Both at Dry-Off in Dairy Cows: Milk Production and Somatic Cell Count Outcomes"

_vetsci, 2022, doi:10.3390/vetsci9100559_

Round 1
Reviewer 1 Report
Introduction
This can be reduced in length. As it is, it contains information that is well-documented and known to people working with mastitis. Possibly, a paragraph about advantages and disadvantages of teat sealants and dry-period treatments might fit better.
The objectives must be very clearly stated in a paragraph on their own.
Procedures
2.1. Herd selection. No details about selection of herds are presented. Please include full relevant details or modify title of subsection.
I noticed that the authors did not perform any bacteriological examination. This is a serious omission and the authors MUST clearly mention that in M & M and justify this significant limitation in a new sub-section in Discussion. Lack of bacteriological examination significantly reduces the value of the work.
Results
Another serious flaw that shows lack of due diligence by the authors. This also hinders full evaluation of the manuscript.
Author Response
Reviewer 2: This can be reduced in length. As it is, it contains information that is well-documented and known to people working with mastitis. Possibly, a paragraph about advantages and disadvantages of teat sealants and dry-period treatments might fit better.
The objectives must be very clearly stated in a paragraph on their own.
AU: We have further edits the introduction to reduce and streamline it as requested.
Reviewer: Procedures
2.1. Herd selection. No details about selection of herds are presented. Please include full relevant details or modify title of subsection.
AU: Herd details added.
Reviewer: I noticed that the authors did not perform any bacteriological examination. This is a serious omission and the authors MUST clearly mention that in M & M and justify this significant limitation in a new sub-section in Discussion. Lack of bacteriological examination significantly reduces the value of the work.
AU: Agreed, but the microbiology is the subject of a different manuscript.
Reviewer: Results
Another serious flaw that shows lack of due diligence by the authors. This also hinders full evaluation of the manuscript.
AU: Something missing in this sentence - no details provided.
Reviewer 2 Report
The study presented is complementary to the one previously published by the authors (2nd reference in the bibliography). It is well designed and has relevant scientific interest.
It is suggested that authors should pay attention to some minor formatting issues (line, 117 and 118; line, 283 to 285; line, 331 to 333) and that the sentences in line 495 to 499 should be removed as they do not present results. significant and are redundant with what is exposed in the results.
Additionally, there are major issues, namely whether current good practices in the use of antibiotics have been respected. Since the study design is in common with the one previously published, the limitations of the 2 studies are not fully consistent. Therefore, the limitations of this study should be reformulated. Bearing in mind, in particular, the way in which the choice of antibiotics was made, which rules and procedures were followed prior to their application and the obvious consequences in the interpretation of the results.
Author Response
Reviewer: The study presented is complementary to the one previously published by the authors (2nd reference in the bibliography). It is well designed and has relevant scientific interest.
AU: Thank you.
Reviewer: It is suggested that authors should pay attention to some minor formatting issues (line, 117 and 118; line, 283 to 285; line, 331 to 333) and that the sentences in line 495 to 499 should be removed as they do not present results. significant and are redundant with what is exposed in the results.
AU: Agreed, we deleted the text that was unrelated to the results maintaining only the summarized findings with no additional extrapolation on uses of our results.
Reviewer: Additionally, there are major issues, namely whether current good practices in the use of antibiotics have been respected. Since the study design is in common with the one previously published, the limitations of the 2 studies are not fully consistent. Therefore, the limitations of this study should be reformulated. Bearing in mind, in particular, the way in which the choice of antibiotics was made, which rules and procedures were followed prior to their application and the obvious consequences in the interpretation of the results.
AU: The study authors did not choose the antibiotics which were used on each dairy. The choice of antibiotic was made by the dairy owner in consult with their herd veterinarian. The study authors did not interfere with this management practice, but rather randomized the herd's choice of antibiotic amongst the dry cows enrolled in the study. In regards to the rules and procedures, the respective drug labels were followed on each dairy for the individual choice of dry-off antibiotic tubes, per the randomization allocation.
Both studies' limitations address the issue of using multiple dry off antibiotics across the dairies. The previous study also addressed limitations with regards to mastitis which was an outcome in that paper. This study researched milk production and SCC therefore this paper's limitations addressed those outcomes. Please let us know if this addresses your concern.
Round 2
Reviewer 1 Report
I still have serious concerns regarding the lack of microbiological results. How can one assess the efficacy of treatment with knowledge of the bacteriological outcomes.
This is a very serious omission of the authors.
Author Response
Reviewer: I still have serious concerns regarding the lack of microbiological results. How can one assess the efficacy of treatment with knowledge of the bacteriological outcomes. This is a very serious omission of the authors.
Authors: Our study includes microbiological data on the milk samples collected from the trial, there is no omission here and its coming in manuscript #3. Just that this paper is concerned with our second objective, that being production and SCC. Otherwise, we agree, its important.